# Parallel Neurological and Cardiac Progression in Hereditary Transthyretin Amyloidosis: An Integrated Clinical and Imaging Study

**DOI:** 10.3390/diagnostics15172143

**Published:** 2025-08-25

**Authors:** Grazia Canciello, Stefano Tozza, Leopoldo Ordine, Brigida Napolitano, Giovanni Palumbo, Mariagiovanna Castiglia, Daniela Pacella, Raffaella Lombardi, Giovanni Esposito, Fiore Manganelli, Maria-Angela Losi

**Affiliations:** 1Department of Advanced Biomedical Sciences, University Federico II, 80131 Naples, Italy; grazia.canciello@unina.it (G.C.); leopoldo.ordine@unina.it (L.O.); brigida.napolitano@unina.it (B.N.); raffaella.lombardi@unina.it (R.L.); espogiov@unina.it (G.E.); 2Department of Neuroscience and Reproductive and Odontostomatological Sciences, University Federico II, 80131 Naples, Italy; stefano.tozza@unina.it (S.T.); giopal@unina.it (G.P.); mgiocas@unina.it (M.C.); fiore.manganelli@unina.it (F.M.); 3Department of Public Health, University Federico II, 80131 Naples, Italy; daniela.pacella@unina.it

**Keywords:** hereditary transthyretin amyloidosis, peripheral neuropathy, global longitudinal strain, neurological staging, diastolic disfunction

## Abstract

**Background:** Hereditary transthyretin amyloidosis (ATTRv) is a rare, autosomal dominant multisystem disease caused by pathogenic variants in the transthyretin (TTR) gene. Although ATTRv is classically categorized into “cardiac” and “neurologic” phenotypes, recent evidence suggests a more complex and overlapping disease spectrum. **Objectives:** This study investigates the relationship between neurological staging and cardiac involvement through an integrated assessment of patients with confirmed TTR mutations. **Methods and Results:** Fifty-eight patients with genetically confirmed ATTRv (41% female, mean age 60 ± 15 years) were evaluated. Genotypes included Phe64Leu, Val30Met, Val122Ile, and others. Patients were stratified by neurological stage: G0 (asymptomatic carriers), G1 (symptomatic but ambulatory), and G2 (requiring walking support). Cardiac assessment included clinical evaluation, echocardiography with tissue Doppler, global longitudinal strain (GLS), and NT-proBNP levels. Cardiac markers worsened with neurological stage. NT-proBNP, left ventricular mass index, maximal wall thickness, and E/E′ ratio increased progressively, while GLS declined (G0: –19%, G1: –14%, G2: –13%; *p* < 0.001), indicating a progressive structural and functional myocardial disease. Ejection fraction remained preserved. Neurological stage independently predicted cardiac dysfunction after age adjustment. **Conclusions**: This is the first study to assess cardiac abnormalities across neurological stages in a well-characterized cohort of ATTRv patients. Cardiac involvement in ATTRv begins early, even in asymptomatic carriers, and progresses with neurological deterioration. GLS and diastolic parameters are sensitive indicators of early myocardial dysfunction, highlighting the need for integrated neurologic and cardiac monitoring in all patients with ATTRv, regardless of initial phenotype.

## 1. Introduction

Hereditary transthyretin amyloidosis (ATTRv) is a rare, autosomal dominant, multisystem disorder caused by pathogenic variants in the transthyretin (TTR) gene [1]. These mutations destabilize the native tetrameric structure of the TTR protein, resulting in dissociation into monomers that misfold and aggregate into amyloid fibrils. Progressive extracellular deposition of these fibrils leads to organ dysfunction, primarily affecting the peripheral nervous system and the heart [2,3,4]. Over 130 pathogenic TTR variants have been identified, contributing to the wide heterogeneity in clinical presentation and disease course observed across different populations and individuals.

Historically, the clinical spectrum of ATTRv has been stratified into distinct phenotypes, neuropathic, cardiac, or mixed, based on the underlying TTR variant. For example, the Val30Met mutation is classically associated with early-onset polyneuropathy, particularly in endemic regions such as Portugal and Japan, while the Val122Ile mutation has been linked predominantly to late-onset cardiomyopathy in Afro-Caribbean and African American populations [2,4]. However, recent evidence increasingly challenges this rigid genotype–phenotype correlation. Several studies have documented that mutations previously associated with a single organ involvement may, in fact, present with a broader multisystem burden. For instance, Val122Ile carriers have been reported to develop significant neurological symptoms, including small fiber neuropathy and autonomic dysfunction, while patients with Val30Met may develop progressive cardiac involvement over time [2,3,4].

Alongside the evolving understanding of its clinical spectrum, the diagnosed prevalence of ATTRv has also increased in recent years. In Italy, for example, national prevalence estimates rose from 4.3 to 6.3 cases per million between 2016 and 2020 [5]. This rise likely reflects a combination of enhanced genetic screening, improved clinical recognition, and the establishment of specialized referral centers equipped with coordinated care pathways. Similar trends have been observed globally, suggesting that ATTRv may be substantially underdiagnosed, especially in non-endemic regions [5,6].

Despite the systemic nature of ATTRv being well recognized [7], most clinical and research efforts have continued to focus on individual organ systems. Neurological and cardiac involvement are often studied separately, with outcomes reported by mutation type or dominant phenotype. This compartmentalized approach limits the ability to fully understand the dynamic and interrelated progression of amyloid pathology across organ systems. Notably, large observational registries such as THAOS (Transthyretin Amyloidosis Outcomes Survey) have provided valuable epidemiological insights but have not systematically addressed the interaction between neurological disease stage and cardiac dysfunction [5]. Consequently, current staging systems may underestimate the cumulative burden of disease and overlook critical windows for intervention.

There is thus a growing need for integrated, mutation-independent models of disease evaluation that can capture the full multisystem complexity of ATTRv. In particular, the use of sensitive cardiac imaging modalities—such as global longitudinal strain (GLS), tissue Doppler indices, and NT-proBNP levels—alongside validated neurological staging systems offers the potential to better delineate disease progression and organ interplay [4,7]. Understanding these parallel trajectories may be essential for refining diagnosis, guiding treatment, and anticipating complications.

We aim to address this knowledge gap by evaluating the concurrent progression of neurological and cardiac dysfunction in patients with ATTRv, independent of genotype or initial clinical presentation. By combining neurological disability staging with advanced echocardiographic markers and cardiac biomarkers, this study seeks to provide a more comprehensive understanding of multisystem involvement in ATTRv and support the development of an integrated framework for clinical evaluation.

## 2. Materials and Methods

*Population*. Fifty-eight consecutive subjects with a confirmed pathogenic TTR mutation (41% women, mean age 60 ± 15 years) were prospectively recruited from March 2023 to February 2025. Patients were followed up at both Neurology and Cardiology Departments of the University Hospital Federico II (Naples, Italy). Fifty-four were first seen at an outpatient neurology clinic for ATTRv, whereas four patients were initially evaluated at the outpatient cardiology clinic for unexplained left ventricular (LV) hypertrophy. At the time of the contemporary neurological and cardiological assessment 31 patients were on treatment. In particular, 12 patients were on patisiran, 8 on tafamidis 20 mg, 5 on vutrisiran, 4 on inoserten and the remaining 2 on tafamidis 61 mg. The protocol has been approved by the locally appointed ethics committee. Informed consent was obtained from all subjects involved in the study.

*Neurological assessment*. Each ATTRv patient underwent a comprehensive clinical assessment by a neurologist [4], which included nerve conduction study (NCS) as previously reported [8]. Patients with no evidence of large fiber neuropathy at NCS, underwent quantitative Sensory Testing (QST) as previously described [9]. The Predicted Age of Disease Onset (PADO) was determined for each family, along with the time to PADO, calculated as Time to PADO = PADO-age at evaluation [10].

Patients were classified according to disability in three groups: (G_0_) indicates asymptomatic subjects carrying pathogenic mutation but without clinical or instrumental evidence of neurological involvement; (G_1_) describes a symptomatic patient with the clinical and/or instrumental involvement of peripheral nervous fibers, able to walk without support; (G_2_) indicates patients who require walking aids or wheelchair dependence [11].

*Cardiological assessment*. ATTRv patients were evaluated with a comprehensive clinical and instrumental assessment. Echocardiographic procedures and measurements were consistent with previous descriptions applied in patients with hypertrophic cardiomyopathy and aortic stenosis [12,13,14]. Briefly, LV wall thickness measured at end-diastole from 2D parasternal long-axis images at three levels, mitral valve leaflet tips, papillary muscle and apical level; maximal wall thickness (MWT), defined as the greatest thickness measured at any LV segment was considered indicative of overt cardiac involvement when ≥13 mm [12,13,14]. Relative wall thickness (RWT), defined as 2 times the posterior wall thickness divided by the LV diastolic diameter [15]. LV mass index (LVMi) was obtained using the ASE/EACVI recommended formula divided by body surface area [15]. LV ejection fraction (LVEF) and left atrial volume (LAV) were calculated using the biplane modified Simpson’s method. LAV was indexed for body surface area (LAVI) [15]. Mitral inflow was analyzed for peak E-wave using pulsed Doppler, whereas tissue Doppler imaging allows a spectral display of mitral annulus velocities at septal and lateral corners [16,17,18]. E’ velocity was measured at both mitral annular corners, and the E/E’ ratio was computed [18]. Global longitudinal strain (GLS) was assessed from apical views, ensuring an optimal frame rate (50–70 frames per second) by narrowing the imaging sector to isolate individual myocardial walls. Speckle-tracking analysis was performed using TomTec software (version 4.4), which identifies and tracks myocardial motion through natural acoustic speckles in greyscale 2D images [16,17]. The endocardium was manually traced, and myocardial motion was evaluated using automated software tracking, with manual verification for quality control. Poor-quality segments were excluded if tracking could not be optimized despite manual correction [16,17].

Statistical analysis. All statistical analyses were performed using IBM SPSS Statistics version 29 (IBM-SPSS, Armonk, NY, USA). Continuous variables were described as mean ± standard deviation. Categorical variables were described as the number of cases with corresponding percentages. To assess differences among the three patient groups (G_0_, G_1_, G_2_), statistical comparisons were performed using analysis of variance (ANOVA) for continuous variables. The χ^2^ test was used to compare categorical variables, with the Monte Carlo simulation to obtain exact *p*-values. To evaluate the relationship between neurological severity and cardiac parameters, multiple linear regression models were used, adjusting for age as a potential confounder. Variables were considered significant when the *p*-value was <0.05.

## 3. Results

A total of 58 patients with genetically confirmed ATTRv were included in the analysis. Based on neurological disability staging, 22 subjects were classified as G_0_, 26 as G_1_, and 10 as G_2_. Table 1 summarizes demographic, genetic, and echocardiographic characteristics across these groups.

Patient age significantly increased across neurological stages (G_0_: 53 ± 17 years; G_1_: 61 ± 13 years; G_2_: 72 ± 6 years; *p* = 0.003), suggesting a progressive nature of the disease with aging. There were no significant differences in sex distribution among the groups (females: G_0_: 45%; G_1_: 46%; G_2_: 20%; *p* = 0.320). Similarly, the age at presentation (PADO) did not differ significantly (Table 1).

The most common mutations in the cohort were Phe64Leu (n = 28, 48%) and Val30Met (n = 20, 34%). Less frequent variants included Val122Ile (n = 4, 7%), Glu54Lys (n = 3, 5%), Ile68Leu (n = 1, 2%), and Lys65Asn (n = 1, 2%). One patient had undergone orthotopic liver transplantation. The distribution of Phe64Leu and Val30Met mutations did not differ significantly across neurological stages (G_0_: 11/8; G_1_: 13/6; G_2_: 4/6; *p* = 0.336), suggesting that neurological severity may not be strictly mutation dependent (Table 1).

Interestingly, all Val122Ile patients were initially evaluated in the cardiology setting, reflecting the expected predominance of cardiac manifestations with this mutation, although subsequent neurological progression warranted reclassification within the G staging system.

Cardiac involvement showed a clear correlation with neurological progression. NT-proBNP levels increased significantly across groups, indicating rising cardiac stress with advancing neurological impairment (Table 1 and Figure 1).

Structural cardiac changes paralleled this trend. LAVI increased progressively, consistent with increased LV filling pressures. Similarly, maximal LV wall thickness was significantly greater in more neurologically impaired patients. The prevalence of concentric hypertrophy, defined as wall thickness ≥13 mm, also rose sharply (G_0_: 9%; G_1_: 39%; G_2_: 70%; *p* = 0.002), indicating a progressive infiltration burden.

LV mass index increased significantly across stages, as did relative wall thickness, further supporting the trend toward a restrictive phenotype in patients with higher neurological disability Table 1 and Figure 1).

The prevalence of an MWT ≥ 13 mm increased as well from G0 to G2 (G0 8%; G_1_ 39%; G_2_ 70%; *p* = 0.002) (Table 1 and Figure 2).

Diastolic dysfunction was evident, as shown by a significantly higher E/E’ ratio from G0 to G2 (G0: 8 ± 3, G_1_: 10 ± 5, G_2_: 15 ± 5; *p* = 0.002). LVEF remained stable among groups (G_0_: 61 ± 8%, G_1_: 59 ± 8%, G_2_: 59 ± 9%; *p* = 0.797), indicating that systolic function was relatively preserved (Table 1, Figure 1). However, GLS showed a progressive decline (G_0_: −19 ± 2%, G_1_: −14 ± 3, G_2_: −13 ± 3%; *p* < 0.001), reflecting worsening subclinical myocardial function (Table 1, Figure 1 and Figure 3).

Since GLS and E/E’ are strictly correlated to age, determinants of GLS and E/E’ were modeled using multiple linear regression adjusted by age and groups, with groups retaining their independent role in determining the differences in GLS and E/E prime (*p* < 0.001).

## 4. Discussion

To our knowledge, this is the first study to correlate standardized neurological staging with advanced cardiac parameters, including GLS and NT-proBNP levels—across a genotype-diverse cohort of patients with ATTRv. By systematically evaluating both neurological and cardiac involvement within the same patient population, this study offers novel insights into the multisystem progression of ATTRv, supporting a model of parallel organ involvement that transcends traditional phenotype-based classifications.

Our findings reveal a significant association between worsening neurological disability and progressive structural and functional cardiac abnormalities, independent of genotype. Specifically, we observed a clear gradient across neurological stages for parameters such as LV wall thickness, LV mass index, relative wall thickness, E/E’ ratio, and NT-proBNP levels, all markers of increasing myocardial involvement. Meanwhile, LV ejection fraction remained preserved across all stages, consistent with the classical “preserved EF” profile of infiltrative cardiomyopathy, but GLS, a sensitive indicator of early systolic dysfunction, showed progressive worsening, even at the earliest neurologic stages.

These results challenge the conventional dichotomous classification of ATTRv into “cardiac” and “neurologic” subtypes, which remains prevalent in clinical guidelines and trial design [5,7,19,20,21,22]. While such categorization may be useful during initial assessment, it fails to account for the overlapping and evolving nature of multisystem involvement. Several recent studies have highlighted the overlapping boundaries between phenotypes [6,21,22], and our findings contribute further by demonstrating that even patients traditionally considered “neurologic” exhibit significant cardiac dysfunction, often subclinical and undiagnosed without advanced imaging.

This paradigm shift is particularly relevant considering our findings regarding Val122Ile mutation carriers. Historically associated with isolated late-onset cardiomyopathy, patients with this variant were frequently referred to cardiology services due to unexplained hypertrophy [22,23,24]. However, systematic evaluation uncovered meaningful neurological impairment, emphasizing the risk of underdiagnosing neuropathy in so-called cardiac variants. This observation aligns with recent multicenter data showing substantial neurologic involvement in Val122Ile carriers [6], reinforcing the idea that rigid genotype-based assumptions can obscure the true disease burden.

Moreover, our study helps address a gap left by large registries, such as the Transthyretin Amyloidosis Outcomes Survey (THAOS), which, despite offering critical epidemiologic and genotypic data [5], have not explicitly examined how neurological and cardiac dysfunction interact over the course of the disease. Our single-center, cross-specialty design, while limited in scale, allowed for a controlled and standardized assessment across both domains. This enabled the identification of a consistent pattern: as neurological disability worsens, echocardiographic and biomarker indicators of cardiac involvement also deteriorate. This parallel progression strongly supports the hypothesis that ATTRv is a systemically progressive disorder with tightly linked multisystem pathology.

Importantly, cardiac dysfunction in ATTRv is frequently under-recognized in neurologically dominant presentations, and vice versa. In our study, even patients in early neurological stages (G_0_–G_1_) showed evidence of increased LV mass, impaired GLS, and elevated NT-proBNP—parameters that would typically indicate significant cardiac amyloid burden [25,26]. These findings suggest that myocardial infiltration often begins earlier than clinical symptoms might suggest, and standard echocardiographic measures such as ejection fraction may fail to detect early disease. Advanced modalities like speckle-tracking echocardiography, tissue Doppler imaging, and biomarker profiling (e.g., NT-proBNP) are thus essential for identifying subclinical cardiac involvement, especially in patients with predominant neurologic symptoms [27,28,29].

This systemic and parallel model of ATTRv progression has important implications for clinical practice. First, all patients with suspected or confirmed ATTRv, regardless of mutation or presenting symptoms, should undergo both neurologic and cardiac evaluations at baseline and during follow-up. Neurologic assessments, including nerve conduction studies and autonomic testing, should be integrated into the diagnostic algorithm for patients with cardiac hypertrophy or heart failure of unclear origin. Conversely, cardiac imaging with strain analysis and biomarkers should be routine in patients presenting with neuropathy, even in the absence of cardiac symptoms.

The increasing prevalence of ATTRv reported in recent years, alongside the earlier identification of mixed phenotypes, further supports this integrated approach [5]. Advances in genetic screening, disease-modifying therapies, and increased awareness have led to earlier diagnoses and a growing population of patients with overlapping features. However, published data may still underrepresent neurologic manifestations due to referral and specialty biases. For example, cardiology-led cohorts may overemphasize cardiac features while underreporting peripheral nerve involvement, and vice versa [5]. A mutation-agnostic, multidisciplinary registry framework is therefore essential to accurately capture the full clinical spectrum and optimize patient care [22].

**Study Limitations.** Several limitations should be noted. First, the cross-sectional design precludes conclusions about causality or the direction of the observed associations. Although our findings suggest parallel progression, longitudinal studies are needed to confirm whether neurological and cardiac dysfunction continue to advance together over time. Second, despite being relatively large for a rare disease, our cohort size limited subgroup analyses, particularly for uncommon mutations such as Glu54Lys and Ile68Leu. Third, although key echocardiographic parameters were adjusted for age, residual confounding from comorbidities or environmental factors cannot be ruled out. Fourth, while echocardiography followed standardized protocols, other imaging methods, such as cardiac magnetic resonance or nuclear scintigraphy, were not systematically available, preventing direct histological correlation with myocardial amyloid burden. In addition, the single-center setting limits generalizability but ensures uniform assessment. Last but not least, we cannot account for the treatment status due to cross-sectional design of our study. This is an important limitation, because patients may also migrate from a primarily neurologic or cardiologic presentation to a mixed phenotype over time [30]. The lack of endomyocardial biopsy, myocardial scintigraphy, and longitudinal follow-up further restricts evaluation of disease progression, and the small number of Val142Ile carriers, who typically show cardiological rather than neurological manifestations, may have introduced selection bias.

## 5. Conclusions

Our findings support a growing body of evidence that ATTRv is a progressive, multisystem disorder characterized by tightly linked neurological and cardiac deterioration, independent of genotype or initial symptom profile. Structural and functional cardiac abnormalities—including increased LV mass, impaired GLS, elevated E/E′ ratios, and rising NT-proBNP—are already detectable at early stages of neurologic impairment and worsen in parallel with disease progression.

This study calls for a paradigm shift away from rigid phenotype or mutation-based labels. A comprehensive, multidisciplinary evaluation strategy—integrating cardiac and neurologic assessments—should become standard in the clinical care of ATTRv patients. This mutation-agnostic approach has the potential to enhance diagnostic accuracy, enable earlier therapeutic intervention, and ultimately improve outcomes for patients facing this complex and often underrecognized multisystem disease.

## Figures and Tables

**Figure 1 diagnostics-15-02143-f001:**
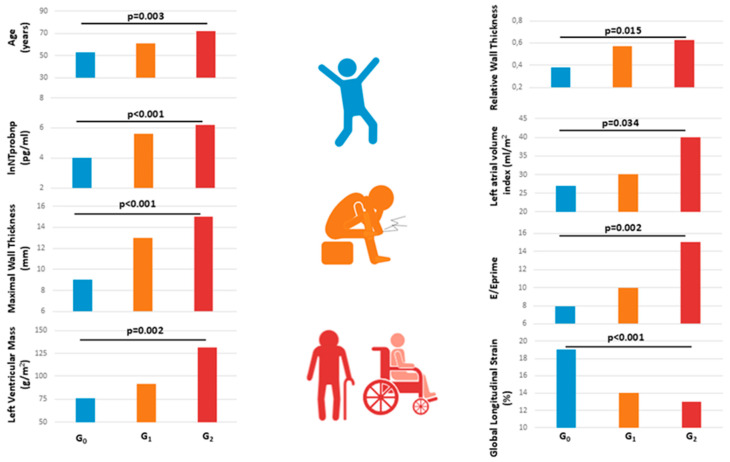
Comparison of cardiac parameters across ATTRv patient groups stratified by neurological severity. G_0_ = Asymptomatic subjects G_1_ = symptomatic patients with the clinical and/or instrumental involvement of peripheral nervous fibers, able to walk without support; G_2_ = patients who need walking aids or wheelchair dependence.

**Figure 2 diagnostics-15-02143-f002:**
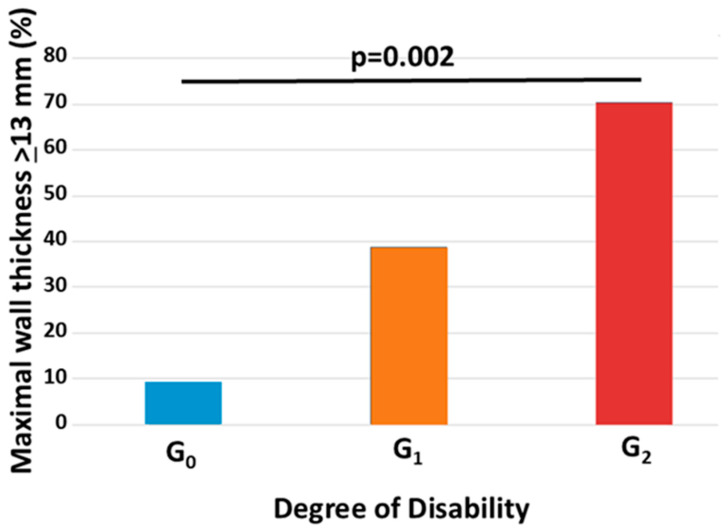
Prevalence among disability groups of maximal wall thickness ≥ 13 mm. G_0_= Asymptomatic subjects G_1_ = symptomatic patients with the clinical and/or instrumental involvement of peripheral nervous fibers, able to walk without support; G_2_ = patients who need walking aids or wheelchair dependence.

**Figure 3 diagnostics-15-02143-f003:**
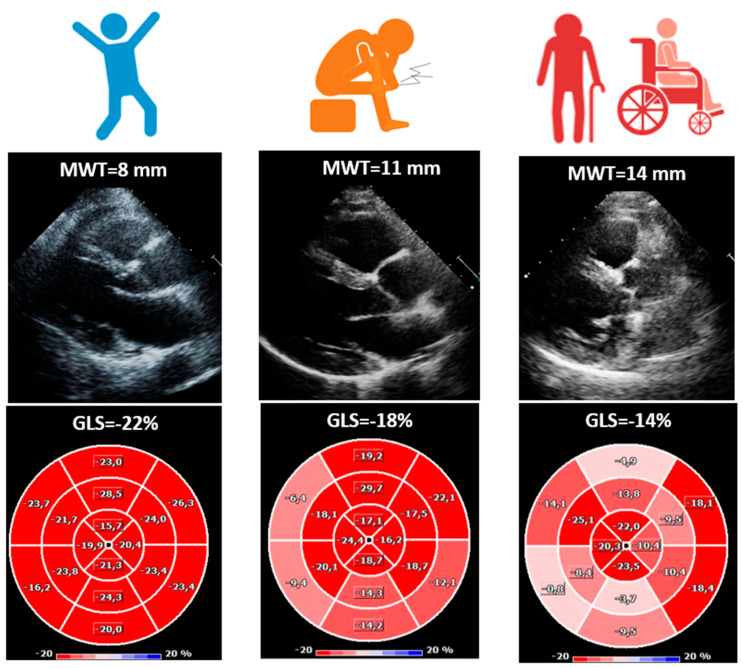
Example of maximal wall thickness and of global longitudinal strain. GLS = global longitudinal strain; MWT = maximal wall thickness. MWT and GLS in a mutation TTR carrier (**left panel**), in a symptomatic patient with ATTRv able to walk without support (**central panel**), and in symptomatic patient with ATTRv which need walking aids (**right panel**).

**Table 1 diagnostics-15-02143-t001:** Demographic and echocardiographic characteristics among ATTRv patients classified by G stages. PADO: predicted age of disease onset.

	Neurological Group			
Variable	G_0_ (n = 22)	G_1_ (n = 26)	G_2_ (n = 10)	*p*
Age (year)	53 ± 17	61 ± 13	72 ± 6	0.003
Sex Female (%)	10	12	2	0.320
PADO (years)	63 ± 8	54 ± 17	65 ± 00	0.143
Mutation Val30Met/Phe64Leu	8/11	6/13	6/4	0.336
lnNT-proBNP (pg/mL)	4.1 ± 1.1	5.6 ± 1.7	6.2 ± 1.1	<0.001
Left atrial Volume (mL/m^2^)	27 ± 9	30 ± 8	40 ± 21	0.034
Maximal Wall Thickness (mm)	9 ± 2	13 ± 4	15 ± 3	<0.001
Maximal Wall Thickness ≥ 13 mm (%)	9	39	70	0.002
Left Ventricular Mass Index (g/kg)	76 ± 23	92 ± 37	131 ± 34	0.002
Relative Wall Thickness	0.38 ± 0.09	0.57 ± 0.03	0.63 ± 0.16	0.015
E/E prime	8 ± 3	10 ± 5	15 ± 5	0.002
Left Ventricular Ejection fraction (%)	61 ± 8	59 ± 8	59 ± 9	0.797
Global Longitudinal Strain (%)	−19 ± 2	−14 ± 3	−13 ± 3	<0.001

## Data Availability

No new data were created due to privacy or ethical restrictions.

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
