# Peer review of "Parallel Neurological and Cardiac Progression in Hereditary Transthyretin Amyloidosis: An Integrated Clinical and Imaging Study"

_diagnostics, 2025, doi:10.3390/diagnostics15172143_

Round 1
Reviewer 1 Report
Comments and Suggestions for Authors
The authors present an assessment of cardiac parameters across neurological stages in a cohort of Italian ATTRv amyloidosis patients. Although the study is rather small, cross-sectional and (as I understand) retrospective, it is well conducted and has a somewhat new approach to it. I agree that ATTRv amyloidosis should be considered a progressive systemic disease, even though there are indeed (early onset ATTRV30M) cases with a more or less pure neuropathic phenotype. The study results are clear and of interest, although not very surprising given the natural history of the disease. I have some suggestions for revision before the manuscript is ready for publication.
- Nomenclature: Please adhere to the latest nomenclature guidelines from the International Society of Amyloidosis and use the same nomenclature for all TTR variants, e.g. Val122Ile or p.Val142Ile.
- Materials and methods: During which time period was the data collected and then analyzed? I presume that the study was done in retrospect?
- Materials and methods: Were any tissue biopsies performed to secure the diagnosis or was the diagnosis based on genotype and clincial findings?
- Results: Did the patients receive any disease-modifying treatment and, in that case, was this taken into account when analyzing the data?
- References: There are older publications (pre-treatment era) on the natural history of ATTRv amyloidosis describing the course of the disease both when it comes to polyneuropathy and cardiomyopathy, please consider adding some of those to the reference list.
Author Response
We thank the reviewer for the suggestions. Please find in Black your comments, in black our answers and in black changes within the manuscript The authors present an assessment of cardiac parameters across neurological stages in a cohort of Italian ATTRv amyloidosis patients. Although the study is rather small, cross-sectional and (as I understand) retrospective, it is well conducted and has a somewhat new approach to it. I agree that ATTRv amyloidosis should be considered a progressive systemic disease, even though there are indeed (early onset ATTRV30M) cases with a more or less pure neuropathic phenotype. The study results are clear and of interest, although not very surprising given the natural history of the disease.
We thank the reviewer for her/his consideration in our work.
I have some suggestions for revision before the manuscript is ready for publication. Nomenclature: Please adhere to the latest nomenclature guidelines from the International Society of Amyloidosis and use the same nomenclature for all TTR variants, e.g. Val122Ile or p.Val142Ile.
You are perfectly right; accordingly, we changed Val142Ile into Val122Ile.
Materials and methods: During which time period was the data collected and then analyzed? I presume that the study was done in retrospect?
Sorry for missing this information. We collected data within 2 years, from march 2023 to February 2025. The study was prospective. These data have been added in the methoid section.
….. were prospectively recruited from March 2023 to February 2025…
Materials and methods: Were any tissue biopsies performed to secure the diagnosis or was the diagnosis based on genotype and clinical findings?
There were no tissue biopsies performed to secure the diagnosis.
Results: Did the patients receive any disease-modifying treatment and, in that case, was this taken into account when analyzing the data?
We thank the reviewer for this interesting point. In fact, 31 patients were already on treatment at the time of the contemporary neurological and cardiological assessment. Unfortunately, due to cross sectional nature of our study, these data cannot be used to understand the role of treatment in the progression of the disease. Anyway, we reported both the presence of treatment and the specific drug in the results section.
….At the time of the contemporary neurological and cardiological assessment 31 patients were on treatment. In particular, 12 patients were on patisiran, 8 on tafamidis 20 mg, 5 on vutrisiran, 4 on inoserten and the remaing 2 on tafamidis 61 mg…..
References: There are older publications (pre-treatment era) on the natural history of ATTRv amyloidosis describing the course of the disease both when it comes to polyneuropathy and cardiomyopathy, please consider adding some of those to the reference list.
We thank the reviewer for this suggestion, which however could not be satisfied, due to limitation of words
Reviewer 2 Report
Comments and Suggestions for Authors
Congratulations to the authors – the article is very well-written and valuable due to its insightful clinical and imaging studies. The study's conclusions are very important.
I have only one comment – the Methods section lacks a detailed discussion of the study population. What was the average follow-up time for the study patients? When were they enrolled in the study? How many people had previously received therapy – or were all enrolled before treatment?
Author Response
Congratulations to the authors – the article is very well-written and valuable due to its insightful clinical and imaging studies. The study's conclusions are very important.
We thank the reviewer for her/his consideration in our study.
I have only one comment – the Methods section lacks a detailed discussion of the study population. What was the average follow-up time for the study patients? When were they enrolled in the study? How many people had previously received therapy – or were all enrolled before treatment?
We thank the reviewer for these interesting points. We answered to this point in the results section. Concerning the follow-up period, this information was not available in our study, in that it was cross-sectional.
….….At the time of the contemporary neurological and cardiological assessment 31 patients were on treatment. In particular, 12 patients were on patisiran, 8 on tafamidis 20 mg, 5 in vutrisiran, 4 in inoserten and the remaing 2 on tafamidis 61 mg…..
Reviewer 3 Report
Comments and Suggestions for Authors
The problem of timely diagnosis of cardiac lesions and proper management of patients with Hereditary transthyretin amyloidosis is of great importance. This study is focused on the comprehensive assessment of both the neurological status of the patient and cardiac lesions, regardless of the region of the gene (cardiopathic or neuropathic) in which the mutation is detected. It is demonstrated that the phenotype of the majority of patients is combined, showing a convincing association between the severity of cardiac lesions and neurologic symptomatology.
The study was performed in a single center, which is partly its limitation, but at the same time it has the advantage that the patients were examined under a single protocol. Endomyocardial biopsy and myocardial scintigraphy data would certainly have embellished the study, but I realize that this may cause technical and financial difficulties. Another important limitation is the lack of follow-up to assess whether cardiac parameters worsen as patients progress through the G0-G1-G2 groups. In addition, there are relatively few patients with the Val142Ile mutation, for which neurologic manifestations with minimal cardiac involvement are typical, which may lead to selection bias.
There are a number of small details that need clarification:
- In the abstract, the sentence (lines 30-32) should be reformulated. The authors write: "Cardiac markers worsened with neurological stage". It is clear. But next sentence says: “NT-proBNP, left ventricular mass index, maximal wall thickness, and E/E′ 31 ratio increased progressively, while GLS declined (G0: -19%, G1: -14%, G2: -13%; 32 p < 0.001)”. But in the full text, the table logically shows that all of these ratios worsen as G0-G1-G2 neurologic symptoms worsen. Clarify the formulation, please.
- The section on materials and methods lacks information on DNA diagnostic process. Also, in the table in the results, it is advisable to add information not only about the number of mutations in Val30Met and Phe64Leu, but also about others, such as “Val30Met/Phe64Leu/other”.
- Line 150 needs to refer to a specific table (Table 1), as well as other places that mention table data. The table itself says Table 0, probably you mean 1? Very strange footnote to the table, it looks more like a text fragment. It would be more reasonable to put the table name and the abbreviations used in the table there.
- In lines 159-161, 167-168 and many times further on in the text, the data from the table are duplicated. Is there a need for this? Perhaps it would be enough to talk about the discovered regularities in a descriptive character? Also, there is duplication of data in the figures.
- The graphical abstract is numbered as Graf 0. Perhaps it is figure 1? Then the figure numbering should be corrected.
Author Response
The problem of timely diagnosis of cardiac lesions and proper management of patients with Hereditary transthyretin amyloidosis is of great importance. This study is focused on the comprehensive assessment of both the neurological status of the patient and cardiac lesions, regardless of the region of the gene (cardiopathic or neuropathic) in which the mutation is detected. It is demonstrated that the phenotype of the majority of patients is combined, showing a convincing association between the severity of cardiac lesions and neurologic symptomatology.
We thank the reviewer for her/his consideration in our work.
The study was performed in a single center, which is partly its limitation, but at the same time it has the advantage that the patients were examined under a single protocol. Endomyocardial biopsy and myocardial scintigraphy data would certainly have embellished the study, but I realize that this may cause technical and financial difficulties. Another important limitation is the lack of follow-up to assess whether cardiac parameters worsen as patients progress through the G0-G1-G2 groups. In addition, there are relatively few patients with the Val142Ile mutation, for which neurologic manifestations with minimal cardiac involvement are typical, which may lead to selection bias.
We thank the reviewer for these important points which now appear in the limitation section.
Several limitations should be noted. First, the cross-sectional design precludes conclu-sions about causality or the direction of the observed associations. Although our find-ings suggest parallel progression, longitudinal studies are needed to confirm whether neurological and cardiac dysfunction continue to advance together over time. Second, despite being relatively large for a rare disease, our cohort size limited subgroup anal-yses, particularly for uncommon mutations such as Glu54Lys and Ile68Leu. Third, alt-hough key echocardiographic parameters were adjusted for age, residual confounding from comorbidities, treatment status, or environmental factors cannot be ruled out. Fourth, while echocardiography followed standardized protocols, other imaging methods, such as cardiac magnetic resonance or nuclear scintigraphy, were not sys-tematically available, preventing direct histological correlation with myocardial amy-loid burden. Finally, the single-center setting limits generalizability but ensures uniform assessment. The lack of endomyocardial biopsy, myocardial scintigraphy, and longitu-dinal follow-up further restricts evaluation of disease progression, and the small number of Val142Ile carriers, who typically show cardiological rather than neurological manifestations, may have introduced selection bias
There are a number of small details that need clarification:
- In the abstract, the sentence (lines 30-32) should be reformulated. The authors write: "Cardiac markers worsened with neurological stage". It is clear. But next sentence says: “NT-proBNP, left ventricular mass index, maximal wall thickness, and E/E′ 31 ratio increased progressively, while GLS declined (G0: -19%, G1: -14%, G2: -13%; 32 p < 0.001)”. But in the full text, the table logically shows that all of these ratios worsen as G0-G1-G2 neurologic symptoms worsen. Clarify the formulation, please.
Thank you for your attention in our paper. We reformulated the sentences, clarifying that all parameters worsen.
NT-proBNP, left ventricular mass index, maximal wall thickness, and E/E′ ratio increased progressively, and GLS declined (G0: -19%, G1: -14%, G2: -13%; 32 p < 0.001), indicating a progressive structural and functional myocardial disease.
- The section on materials and methods lacks information on DNA diagnostic process. Also, in the table in the results, it is advisable to add information not only about the number of mutations in Val30Met and Phe64Leu, but also about others, such as “Val30Met/Phe64Leu/other”.
We thank the reviewer for this point. To unchange the readability of the table, we added this information in the text.
The most common mutations in the cohort were Phe64Leu (n=28, 48%) and Val30Met (n=20, 34%). Less frequent variants included Val122Ile (n=4, 7%), Glu54Lys (n=3, 5%), Ile68Leu (n=1, 2%), and Lys65Asn (n=1, 2%). One patient had undergone orthotopic liver transplantation.
- Line 150 needs to refer to a specific table (Table 1), as well as other places that mention table data. The table itself says Table 0, probably you mean 1? Very strange footnote to the table, it looks more like a text fragment. It would be more reasonable to put the table name and the abbreviations used in the table there.
Thank you. You are perfectly right. We renamed our table, which now is Table 1 and reported more correctly the footnote
- In lines 159-161, 167-168 and many times further on in the text, the data from the table are duplicated. Is there a need for this? Perhaps it would be enough to talk about the discovered regularities in a descriptive character? Also, there is duplication of data in the figures.
Again, we would like to thank the reviewer for her/his attention in our paper. We recognized that there was a redundancy. Thus, we deleted the duplication of data within the manuscript. We decided to keep the data figure, as figures generally increase the likelihood of being cited.
- The graphical abstract is numbered as Graf 0. Perhaps it is figure 1? Then the figure numbering should be corrected.
Thank you. We renumbered our figures, with graphical abstract being figure 1.
Round 2
Reviewer 1 Report
Comments and Suggestions for Authors
The authors have responded to the questions and revised the manuscript accordingly. However, I do believe that the fact that the treatment status of the included patients was not taken into account in the analyses must be mentioned as a limitation of the study. Further, there is in fact a publication from THAOS analysing the "Clinical and Genotype Characteristics and Symptom Migration in Patients With Mixed Phenotype Transthyretin Amyloidosis from the Transthyretin Amyloidosis Outcomes Survey", which should be included as a reference.
Author Response
The authors have responded to the questions and revised the manuscript accordingly.
Thank you for your consideration in our paper.
However, I do believe that the fact that the treatment status of the included patients was not taken into account in the analyses must be mentioned as a limitation of the study. Further, there is in fact a publication from THAOS analysing the "Clinical and Genotype Characteristics and Symptom Migration in Patients With Mixed Phenotype Transthyretin Amyloidosis from the Transthyretin Amyloidosis Outcomes Survey", which should be included as a reference.
Thank you again for your valuable suggestion. We have clarified this limitation more explicitly in the study’s limitation section. In addition, we have included the mentioned work and apologize for previously omitting this important publication
Study Limitations. Several limitations should be noted. First, the cross-sectional design precludes conclusions about causality or the direction of the observed associations. Although our findings suggest parallel progression, longitudinal studies are needed to confirm whether neurological and cardiac dysfunction continue to advance together over time. Second, despite being relatively large for a rare disease, our cohort size limited subgroup analyses, particularly for uncommon mutations such as Glu54Lys and Ile68Leu. Third, although key echocardiographic parameters were adjusted for age, residual confounding from comorbidities or environmental factors cannot be ruled out. Fourth, while echocardiography followed standardized protocols, other imaging methods, such as cardiac magnetic resonance or nuclear scintigraphy, were not systematically available, preventing direct histological correlation with myocardial amyloid burden. In addition, the single-center setting limits generalizability but ensures uniform assessment. Last but not least, we cannot account for the treatment status due to cross sectional design of our study. This is an important limitation, because patients may also migrate from a primarily neurologic or cardiologic presentation to a mixed phenotype over time [31] The lack of endomyocardial biopsy, myocardial scintigraphy, and longitudinal follow-up further restricts evaluation of disease progression, and the small number of Val142Ile carriers, who typically show cardiological rather than neurological manifestations, may have introduced selection bias.
Reviewer 3 Report
Comments and Suggestions for Authors
The authors have done serious work to improve the manuscript. In its current form, the paper is much more structured, the clarifications in limitations section remove the questions that previously arose when reading the results. I thank the authors for their attention to my comments and wish them success in their future research.
Author Response
We sincerely appreciate your kind words regarding our revised paper